# Research on Self-Noise Suppression of Marine Acoustic Sensor Arrays

**Haoyu Tan** †, **Guochang Liu** †, **Haoxuan Li, Guojun Zhang, Jiangong Cui**, **Yuhua Yang, Changde He**, **Licheng Jia, Wendong Zhang** and **Renxin Wang** *

State Key Laboratory of Dynamic Measurement Technology, North University of China, Taiyuan 030051, China
* Correspondence: wangrenxin@nuc.edu.cn
† These authors contributed equally to this work.

**Abstract:** Marine acoustic sensors can detect underwater acoustic information. The cilium micro-electro-mechanical system (MEMS) vector hydrophone (CVH) is the core component of the ocean noise measurement system. The performance of the CVH, especially its self-noise, has received widespread attention. In this paper, we propose a solution to improve the performance of the CVH using an array to detect environmental noise in a complex deep-water environment. We analyzed the self-noise source of the CVH and the noise suppression principle of the four-unit MEMS vector hydrophone (FUVH). In addition, we designed the pre-circuit of the FUVH, completed the cross-beam structure by the MEMS processing, and packaged a FUVH. Then, we tested the performance of a packaged FUVH. Finally, the experimental results show that the FUVH reduces the self-noise voltage power spectrum by 6 dB compared to the CVH structure. The FUVH achieves better linearity at low frequencies without reducing the bandwidth and sensitivity. In addition, it minimizes the equivalent self-noise levels by 5.18 and 5.14 dB in the X and Y channels, respectively.

**Keywords:** MEMS; ocean ambient noise; acoustic sensor; vector hydrophone; array; self-noise suppression

## 1. Introduction

Ocean ambient noise is generated by wind and waves, rainfall, ships, marine organisms, human industrial activities, and other factors [1]. All kinds of measuring equipment, such as sonar systems, unmanned underwater vehicles, and air-dropped sonar buoys located in the ocean, are disturbed by the noise of the ocean environment, which limits their performance. Therefore, evaluating ocean ambient noise is an effective way to acquire ocean information. A buoy system is usually designed to measure and record ocean ambient data. Figure 1 shows a common ocean ambient noise measurement system realized by a buoy, in which a vector hydrophone is the core component [2]. The buoy falls at a constant speed after being released from a vessel. It collects and records ocean ambient noise data when it reaches the deep sea. After ejecting the load weight, the ocean ambient noise measurement system can automatically rise to the sea surface. The data can be transmitted to planes or satellites. After processing, the ocean ambient noise data can be applied to optimize other equipment data or realize underwater environment monitoring [3].

The vector hydrophone can detect underwater vector information. So, the vector hydrophone-based ocean noise measurement system can identify the direction of the noise source [4]. In recent years, the micro-electro-mechanical system (MEMS) process has enabled the structure of vector hydrophones to reach the micron level. The advantage of the smaller size is reflected in many aspects, such as lower cost, lower power consumption, higher reliability, and easier integration. The smaller volume for the ocean noise measurement system makes the array (linear array, circular array, etc.) more accessible.

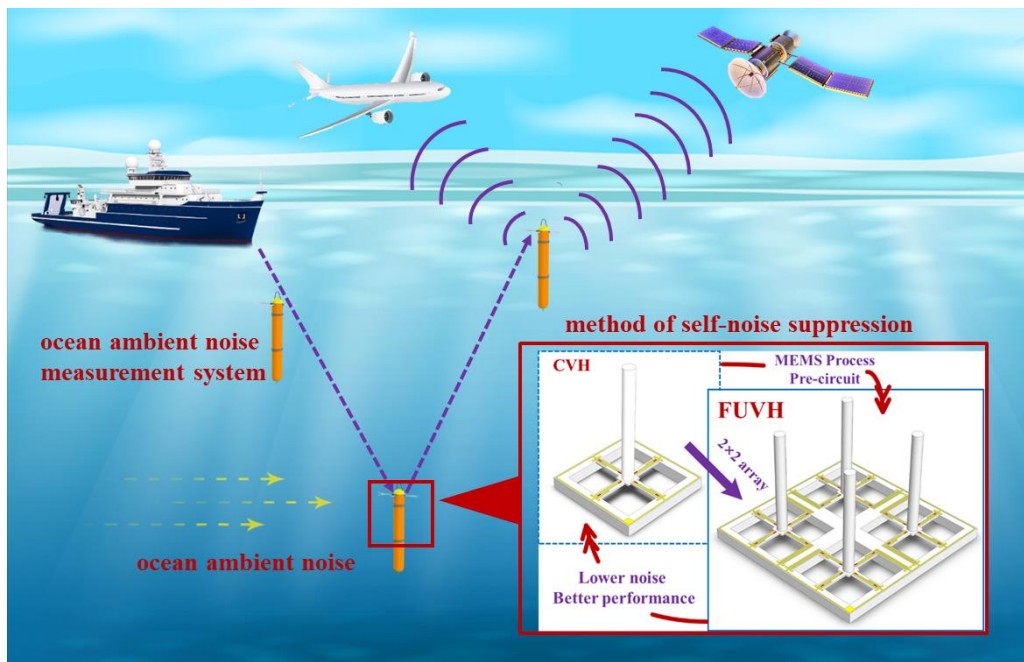

**Figure 1.** Working sketch of ocean ambient noise measurement system.

Detecting weak and complex noise or signals in an underwater environment is the biggest challenge for MEMS vector hydrophones today [5]. Since its inception, the cilium MEMS vector hydrophone (CVH) has been continuously improved by increasing the sensor's performance. By changing the shape of the cilium, some approaches extend the sensing area to increase the device's sensitivity, whereas others increase the number of measurement dimensions to enable three-dimensional measurement of underwater information [6–11]. Alternatively, by optimizing the cross-beam, some increase the stress concentration area to achieve high sensitivity measurement [12], and others increase the cantilever beam to achieve three-dimensional measurement [13]. The above schemes are all based on the improvement of a single CVH. At the same time, in 2016, Liu proposed expanding the CVH through the MEMS process into a four-unit MEMS vector hydrophone (FUVH) array, where he integrated different lengths of cilia to achieve broadband measurements (20–5000 Hz) [14]. In 2019, Zhang theoretically derived the correlation coefficients of the individual array units of the FUVH, and the conclusion showed that the four-unit array could ensure a high degree of output consistency at the current structure size and operating bandwidth [15]. After that, in 2021, Zhang found that the integration of rigid ring cilia on the FUVH obtained higher sensitivity than a single CVH [16].

However, the sensor used in the ocean noise measurement system needs to reduce the self-noise level. Because we can not detect ocean ambient noise levels that are lower than the self-noise level, suppressing noise from the structure or pre-circuit is our target. In 2019, Petrov reported using a combined receiver instead of a conventional acoustic pressure hydrophone to improve the signal-to-noise ratio (SNR) [17]. The combined receivers consist of a hydrophone and a three-component vector receiver. This scheme used complex algorithms to achieve a high SNR ratio. In 2020, Tang suppressed thermal and acoustic imaging noise utilizing an additive circuit with multichannel amplifiers, and experimentally verified a 5.547 dB improvement in SNR [18]. In the same year, Yang prepared a $10 \times 10$ array MEMS hydrophone based on an AlN piezoelectric film and obtained lower noise than that of most previously reported MEMS hydrophones [19]. However, this paper only considered choosing a suitable piezoelectric material, and optimizing the amplification circuit and matching layer, and did not explain noise reduction from the direction of the array.

In this study, to optimize the performance of the ocean ambient noise measurement system, we proposed the suppression of the self-noise of the cilia-based MEMS vector

hydrophone using FUVH. We analyzed the noise sources and the principle of suppression. Then, we verified the approach experimentally. The FUVH has good performance in terms of lower noise and vector detection with moderate sensitivity.

## 2. Sensitivity Principle

The chip of the CVH consists mainly of the sensitive cilium and a cross-beam, the structure of which is shown in Figure 2. When an acoustic transmission passes through the hydrophone, the sensitive cilium generates a moment due to the acoustic pressure difference, which deflects the support block. Because the cross-beam and the support block are integrated, the deflection of the support block produces deformation in the beam. The stress at any point on the beam is [20]:

$$\sigma_{(x)} = \pm \frac{L^2 + 3aL - 3x(a+L)}{\frac{2}{3}bt^2(L^2 + 3aL + 3a^2)}(F_x \cdot H) \tag{1}$$

where $L$ is the length of the beam, $a$ is half the length of the support block's side, $b$ is the width of the beam, $t$ is the thickness of the beam, $F_x$ is the force generated by the sound pressure on the cilium, and $H$ is the height of the cilium.

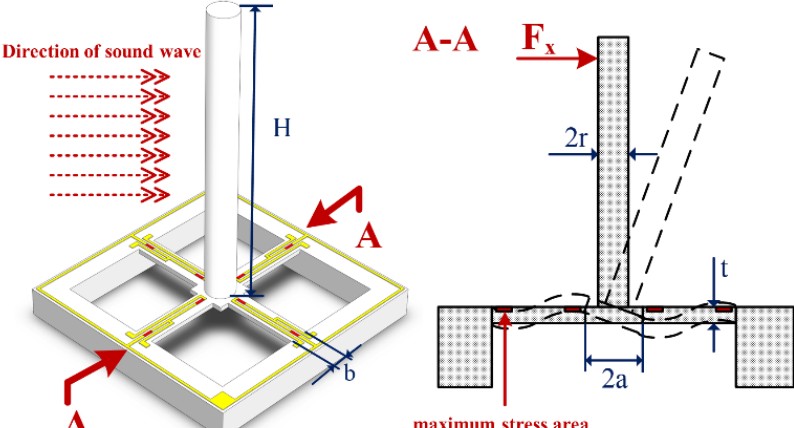

**Figure 2.** The CVH structure. The acoustic wave generates a force Fx on the cilium, which causes the mass block to drive the beam into deformation. The maximum stresses generated by the deformation are located at the ends of the beam.

The FUVH is shown in Figure 3. Piezoresistors are distributed at the root of the beam to obtain the maximum stress and convert the beam's stress change into resistance change. The difference in resistance is:

$$\frac{\Delta R}{R} = \pi \cdot \sigma_{max} \tag{2}$$

where $\pi$ is the piezoresistive coefficient, $\sigma_{max}$ is the maximum stress.

To output the voltage variation rather than the resistance variation, we connect piezoresistors to the Wheatstone bridge. The CVH has two sets of Wheatstone bridges for the X and Y channels, whereas the FUVH has Wheatstone bridges in eight locations. The voltage output of a single channel is:

$$U_x = U_o \left( \frac{R_1 + \Delta R_1}{R_1 + \Delta R_1 + R_2 - \Delta R_2} - \frac{R_4 - \Delta R_4}{R_3 + \Delta R_3 + R_4 - \Delta R_4} \right) \tag{3}$$

where $U_o$ is the bridge input voltage.

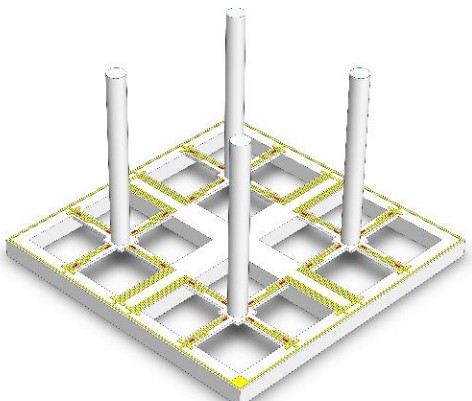

**Figure 3.** The FUVH structure. The piezoresistors are located at the root of the cross-beam and wired as a Wheatstone bridge.

In the ideal case, the resistance is equal, and the stress at the corresponding position on the beam is equal. Substituting (2) into (3), the voltage output of a single channel is:

$$U_X = U_0 \cdot \pi \cdot \sigma_{max}. \tag{4}$$

Then, we can obtain the orientation information of the underwater target by processing the data of X and Y channels.

### 3. Noise Analysis

*3.1. Electrical Noise*

The pre-circuit of the sensor is an essential part of signal acquisition. Minimizing the electrical noise of the circuit is a vital aspect to be considered in the design of the circuit.

The pre-circuit of the CVH consists of a Wheatstone bridge, an instrumentation amplifier circuit, and a filter circuit. The instrumentation amplifier circuit provides high gain, and the filter circuit limits the bandwidth of the circuit. The first-stage circuit with high gain is usually the primary noise source for multi-stage circuits. The equivalent noise source circuit for the instrumentation amplifier circuit is shown in Figure 4.

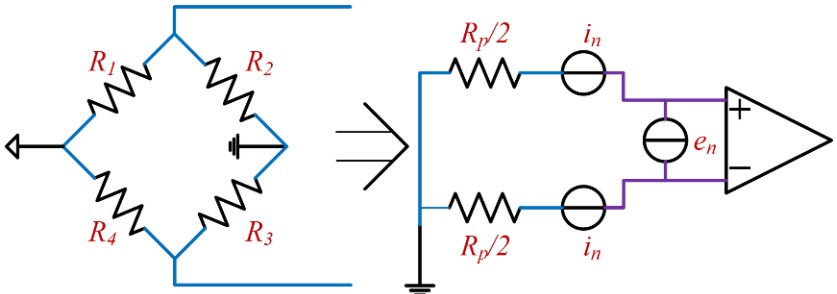

In the ideal condition, $R_1 = R_2 = R_3 = R_4 = R_p$

**Figure 4.** Equivalent noise source circuit. In the ideal condition, the resistance is equal and is $R_p$. The four resistors of the Wheatstone bridge are equivalent to two resistors of value $R_p/2$ at both ends of the amplifier.

The resistance thermal noise voltage density generated by the piezoresistors is

$$D_{R+} = D_{R-} = \sqrt{4 k_B T \frac{R_p}{2}} \tag{5}$$

where $R_p$ is the zero-input resistance of the piezoresistors, and the equivalent resistance of the Wheatstone bridge at the positive and negative ends of the amplifier is $R_p/2$. The current noise source of the amplifier converts to voltage noise through the equivalent resistance, and the voltage density is:

$$D_{I+} = D_{I-} = i_n \frac{R_P}{2} \tag{6}$$

where $i_n$ is the amplifier input equivalent current noise density. Then, the total output noise voltage density is:

$$D_{out} = A_v \cdot D_{in} = G\sqrt{D_{R+}^2 + D_{R-}^2 + D_{I+}^2 + D_{I-}^2 + e_{ni}^2 + (e_{no}/G)^2} \tag{7}$$

where $A_v$ is the amplifier amplification, $e_{ni}$ is the input voltage noise density of the amplifier, and $e_{no}$ is the output voltage noise density. This value characterizes the electrical noise of the sensor in the frequency spectrum of the white noise density, which usually determines the total output noise. On the contrary, flicker noise at low frequencies does not influence the total output noise, because its bandwidth is too narrow compared to the operating frequency band of the hydrophone [21].

*3.2. Mechanical-Thermal Noise*

Electrical noise is usually a critical noise source that makes up the sensor's self-noise and determines the sensor's noise floor. However, for microsensors, mechanical-thermal noise may set an even higher noise floor [22,23].

MEMS vector hydrophones feature silicon microstructure that are advantageous in terms of their structural dimensions and sensitivity. The characters make MEMS vector hydrophones more susceptible to mechanical-thermal noise generated by molecular thermal disturbance [24]. The spectral density of the mechanical-thermal noise source is given by the Nyquist relation [25]:

$$F_n = \sqrt{4k_B TR} \tag{8}$$

where $F_n$ is the spectral density of the mechanical-thermal noise source in N/$\sqrt{\text{Hz}}$, $k_B$ is the Boltzmann constant in J/K, $T$ is the absolute temperature in K, and $R$ is the damping constant of the system in N·m/s.

Equation (8) shows that the mechanical-thermal noise is only related to the ambient temperature and system damping. The damping is a mechanism of energy dissipation, which, crucially, includes acoustic radiation, anchor losses, thermoelastic damping, and viscous damping [26]. For cilia-based devices, the hydrodynamic interaction between the cilia and the fluid (silicon oil in FUVH or CVH) shows viscous damping [27]. This depends not only on the sensor structure but also on the fluid properties. Anchor losses can be alleviated by mechanically isolating, and thermoelastic damping as intrinsic damping relates to material and structure. According to the fluctuation–dissipation theorem, the system damping gives a path that allows energy to leave a system and introduces mechanical-thermal noise into this system [23]. The spectral density of mechanical-thermal noise is a white noise spectrum. The system transfer function shapes the spectral density at the sensor output. For CVH or FUVH, we can set out the differential equations of motion for second-order systems [28] as:

$$\ddot{y} + 2\zeta\omega_0\dot{y} + \omega_0^2 y = \frac{x}{m} \tag{9}$$

where $x$ is the system input, $y$ is the system output, $x$ and $y$ are both as a function of displacement with respect to time, $m$ is system equivalent mass, $\omega_0$ is the system natural frequency, and $\zeta$ is the system damping ratio, expressed as:

$$\zeta = \frac{R}{2m\omega_0} \tag{10}$$

Equation (9) can be written as:

$$\left(s^2 + 2\zeta\omega_0 s + \omega_0^2\right)Y = \frac{X}{m} \tag{11}$$

$$H(s) = \frac{Y}{X} = \frac{1}{\left(s^2 + 2\zeta\omega_0 s + \omega_0^2\right)m} = \frac{1}{ms^2 + Rs + \omega_0^2 m} \tag{12}$$

where *H(s)* is the sensor system transfer function. The input displacement spectrum density is [24]:

$$I_{MT} = \frac{\sqrt{4k_B TR}}{m\omega_0^2} \tag{13}$$

the output displacement spectrum density is:

$$O_{MT} = H(s)\cdot I_{MT}. \tag{14}$$

This output is reflected in the movement of the cilia and beam, and is converted to a voltage by piezoresistors. The output voltage spectrum density is:

$$D_{MT} = n\cdot O_{MT} \tag{15}$$

where *n* is a constant related to the parameters of the cilia and beam, the piezoresistive coefficient, and the supplier of Wheatstone bridges. As a result, the shape of $D_{MT}$ depends on *H(s)*. The typical transfer function shape of CVH or FUVH is shown in Figure 5 [28].

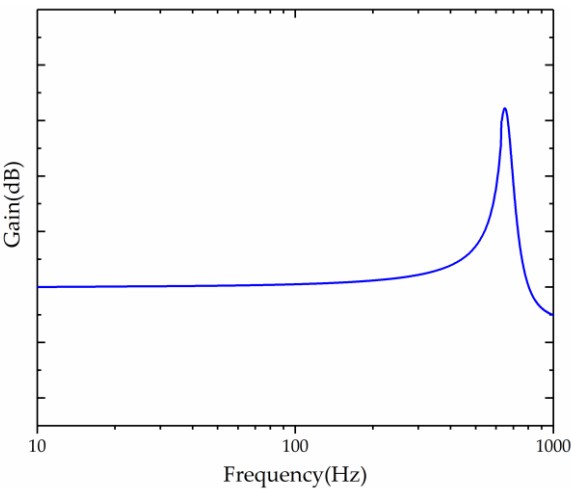

**Figure 5.** The typical transfer function shape of CVH or FUVH.

As shown in Figure 5, the peak is caused by the sensor's resonant frequency. The curve is almost horizontal, like white noise in the working area.

### 3.3. Noise Suppression by FUVH

According to the analysis above, the components of the total output noise of a CVH are white noise and flicker noise, where the effect of white noise is much greater than that of flicker noise on the total noise. These white noises include the electrical circuit's thermal noise and the sensor structure's mechanical-thermal noise.

As shown in Figure 6, suppose a CVH receives a sound signal and ideally outputs a signal of amplitude $V_0$. However, the actual output signal consists of white noise $V_{N\_wh}$ and flicker noise $V_{N\_f}$. That signal is then amplified and filtered by the pre-circuit. At the

same time, the white noise $V_{NC\_wh}$ and flicker noise $V_{NC\_f}$ of the pre-circuit also add to the signal. The output signal is:

$$V_{out} = A_v \cdot V_0. \tag{16}$$

and the output noise is:

$$V_{N\_out} = \sqrt{(Av \cdot V_{N\_wh})^2 + \left(Av \cdot V_{N\_f}\right)^2 + V_{NC\_wh}{}^2 + V_{NC\_f}{}^2}. \tag{17}$$

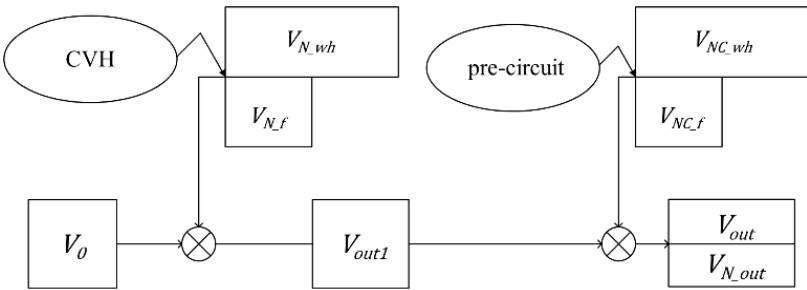

**Figure 6.** Signal transmission process of a single CVH.

Since the contribution of thermal noise is much larger than that of flicker noise over the whole frequency band in which the sensor operates, $V_{N\_out}$ can be approximately equal to:

$$V_{N\_out} \approx \sqrt{(Av \cdot V_{N\_wh})^2 + V_{NC\_wh}{}^2}. \tag{18}$$

Then, substitute the noise spectral density from (7) and (15):

$$V_{NC\_wh} = D_{out}\sqrt{f_{th}} \tag{19}$$

$$V_{N\_wh} = D_{MT}\sqrt{f_{th}} \tag{20}$$

where $f_{th}$ is the system bandwidth, which is often determined by the low-pass filter at the last stage of the circuit.

If the output signals of two CVHs detecting at the same position are superimposed by an adder, as shown in Figure 7, we can obtain:

$$V_{out} = \frac{V_{out1} + V_{out2}}{2} = A_v \cdot V_0 \tag{21}$$

where the result is equal to that of (16). The output noise in this case is:

$$V_{N\_out} = \frac{\sqrt{(Av \cdot V_{N1\_wh} + Av \cdot V_{N2\_wh})^2 + (V_{NC1\_wh} + V_{NC2\_wh})^2}}{2}. \tag{22}$$

Since the white noise is irrelevant at any two moments [29], the output noise can be written as:

$$V_{N\_out} = \frac{\sqrt{(Av \cdot V_{N1\_wh})^2 + (Av \cdot V_{N2\_wh})^2 + (V_{NC1\_wh})^2 + (V_{NC2\_wh})^2}}{2}. \tag{23}$$

If the two CVHs are the same and the pre-circuit devices are identical, then there are:

$$V_{N1\_wh} = V_{N2\_wh} \tag{24}$$

$$V_{NC1\_wh} = V_{NC2\_wh}. \tag{25}$$

Ideally, the output noise would be:

$$V_{N\_out} = \frac{\sqrt{(Av \cdot V_{N\_wh})^2 + (V_{NC\_wh})^2}}{\sqrt{2}}. \tag{26}$$

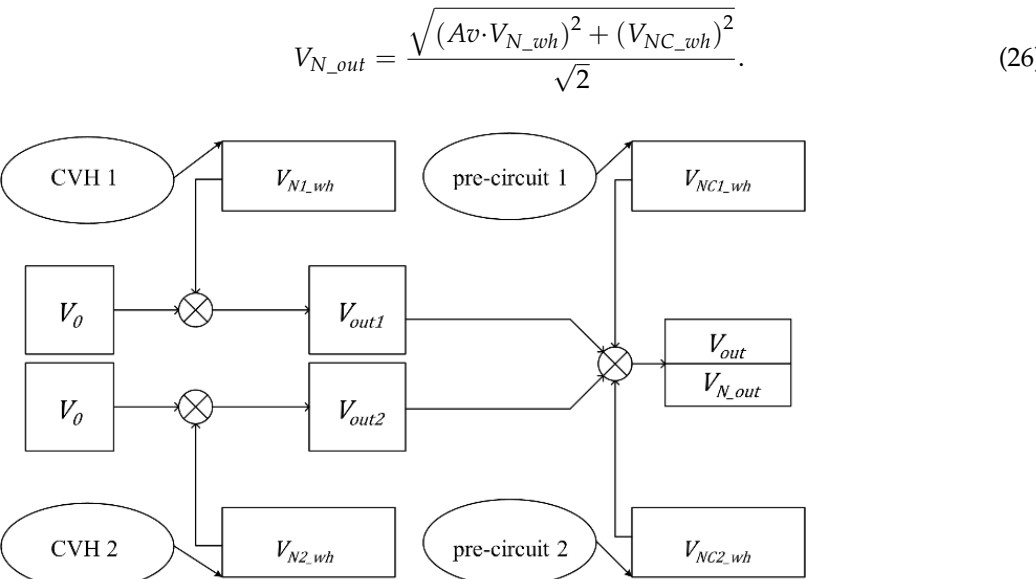

**Figure 7.** Signal transmission process of two CVHs.

So, ideally, the superposition of two CVHs by an adder reduces the noise by 3 dB compared to a single CVH, when they have a consistent output signal voltage. Similarly, four superpositions can reduce the noise by 6 dB and keep the output signal voltage constant. We designed the pre-circuit based on the above analysis, and the circuit block diagram of the FUVH is therefore shown in Figure 8.

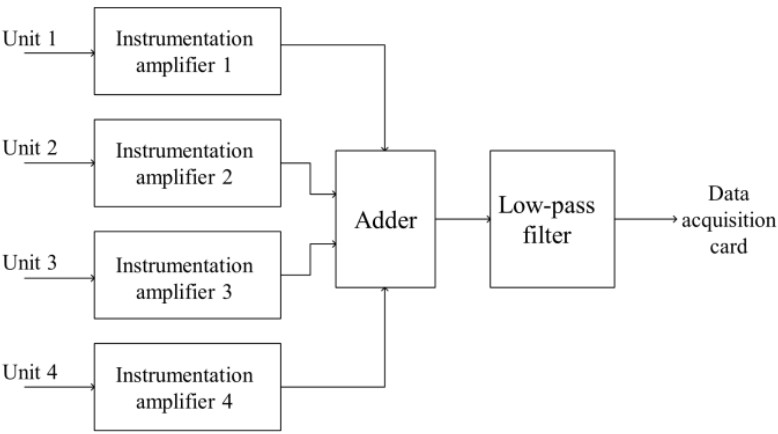

**Figure 8.** The circuit block diagram of the FUVH.

## 4. Experimental Program

To test the actual effect of FUVH on noise suppression, we fabricated a FUVH. Then, we prepared the cross-beam structure of the hydrophone, and formed it into a 2 × 2 array by the MEMS process. The process flow is shown in Figure 9.

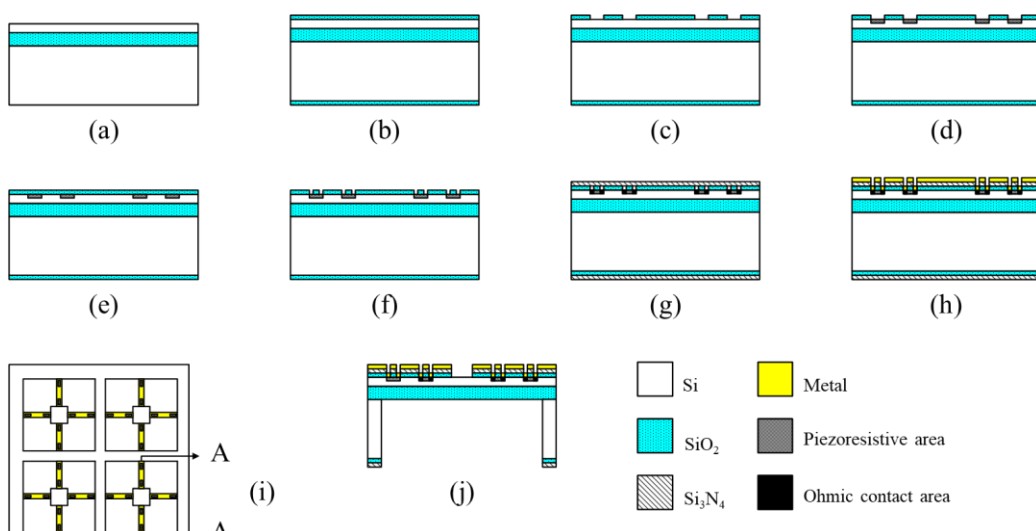

**Figure 9.** The MEMS process of the FUVH (**a**–**h**) and (**j**) the view of the A-A section; (**i**) the top view.

The specific steps are as follows: (a) prepare a 4-inch SOI wafer; (b) double-sided oxide the SOI; (c) RIE etch the oxide layer and carve out the piezoresistors; (d) use low boron ion implantation to form piezoresistors and anneal repair the damaged lattice; (e) use reoxidation; (f) ICP etch out the ohmic contact area; (g) use high boron ion implantation to form an ohmic contact, and anneal and double-sided deposit the silicon nitride; (h) ICP etch the lead hole, and metal sputter to form conductivity; (i) frontal release the structure; (j) backside release the structure. The completed chip is shown in Figure 10.

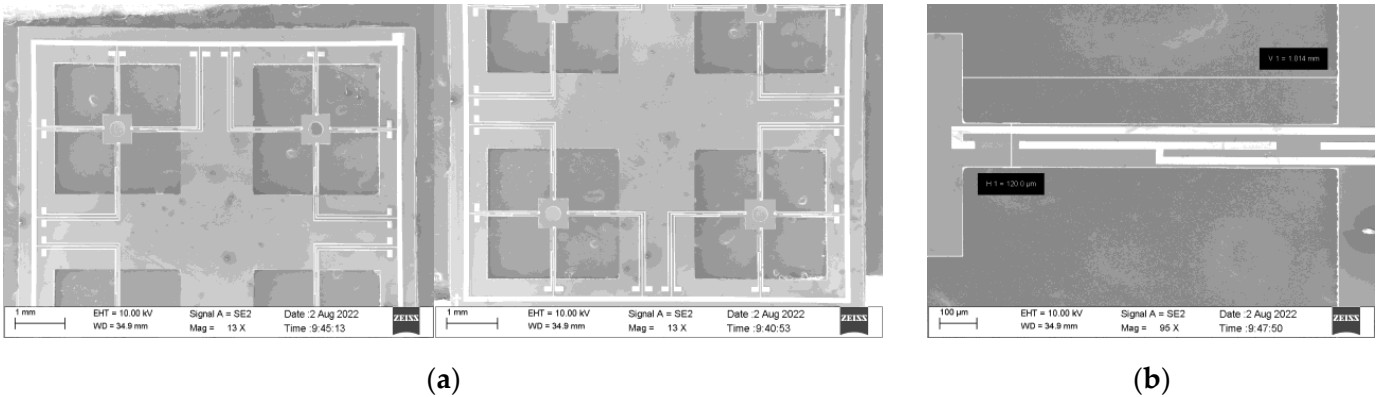

**Figure 10.** SEM image of the completed chip: (**a**) the 2 × 2 array and (**b**) the beam.

The main parameters of FUVH are given in Table 1.

**Table 1.** The main parameters of FUVH.

| Parameter | Description | Value (µm) |
| --- | --- | --- |
| beam length | $L$ | 1000 |
| support block's side length | $2a$ | 600 |
| beam width | $b$ | 120 |
| beam thickness | $t$ | 40 |
| cilium height | $H$ | 8000 |

Then, the processed hydrophone cross-beam chip is integrated with the same lengths of cilia [30]. Next, the integrated chip is oil-filled and packaged. Finally, the completed FUVH is shown in Figure 11.

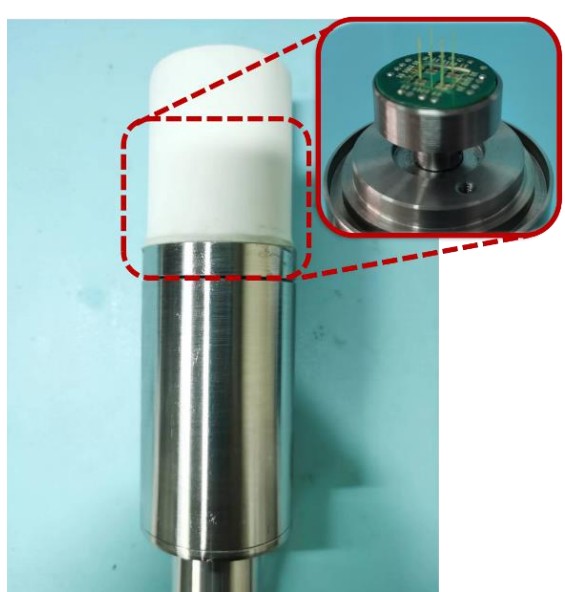

**Figure 11.** Image of the completed FUVH hydrophone.

*4.1. Sensitivity Test*

As shown in Figure 12, the vector hydrophone test device includes a revolver, a calibration tube, a signal generator, a power amplifier, an emission transducer, and a data acquisition card. The FUVH is elastically suspended from the revolver to reduce the impact of platform vibration [31]. Then, the FUVH is placed at the same liquid level as the reference standard hydrophone to simplify calculation [32].

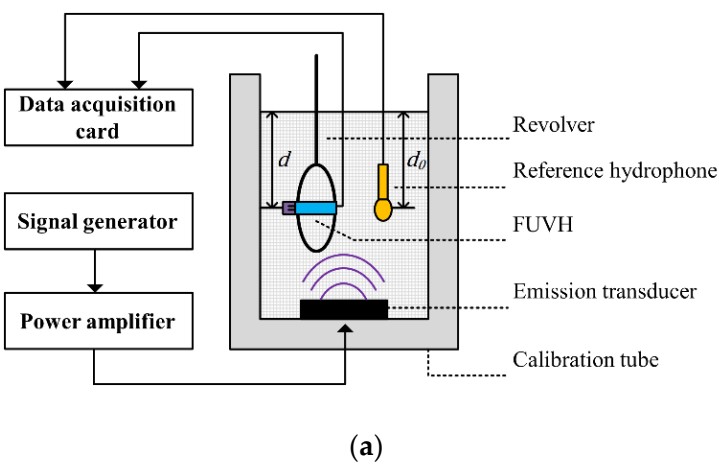
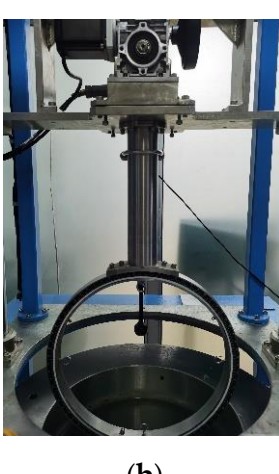

(**a**) (**b**)

**Figure 12.** *Cont.*

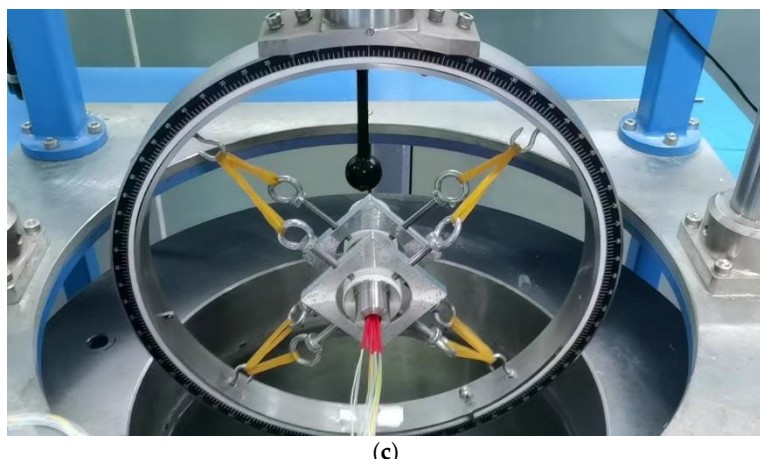

(**c**)

**Figure 12.** The vector hydrophone test device: (**a**) the FUVH distance from the water surface is *d*, and the reference hydrophone distance from the water surface is $d_0$; (**b**) the test device and (**c**) the FUVH elastically suspended from the revolver.

We used the revolver rotating the FUVH in the X and Y directions, and sensitivity tests were carried out in both directions using the comparative calibration method [33]. The emission transducer emitted sound from 10 to 2000 Hz in 1/3 octave steps. We recorded the output voltage values of the FUVH and its four units using the data acquisition card. At the same time, we recorded the output voltage values of the standard sound pressure hydrophone. Finally, we obtained the transducer sensitivity from the following equation:

$$M_x = 20\lg\left(\frac{e_x}{e_0}\frac{sinkd_0}{coskd}\right) + M_0 \tag{27}$$

where $M_0$ is the sound pressure sensitivity of the standard sound pressure hydrophone in the free field, and the value is −170 dB. $e_0$ and $e_x$ are the open circuit output voltage of the standard sound pressure hydrophone and the FUVH, respectively. $d$ and $d_0$ are the depth of the center of the FUVH and standard hydrophone from the water surface, respectively. In the actual test, we usually place the FUVH and the standard sound pressure hydrophone on the same level, $d = d_0$. The results are shown in Figure 13.

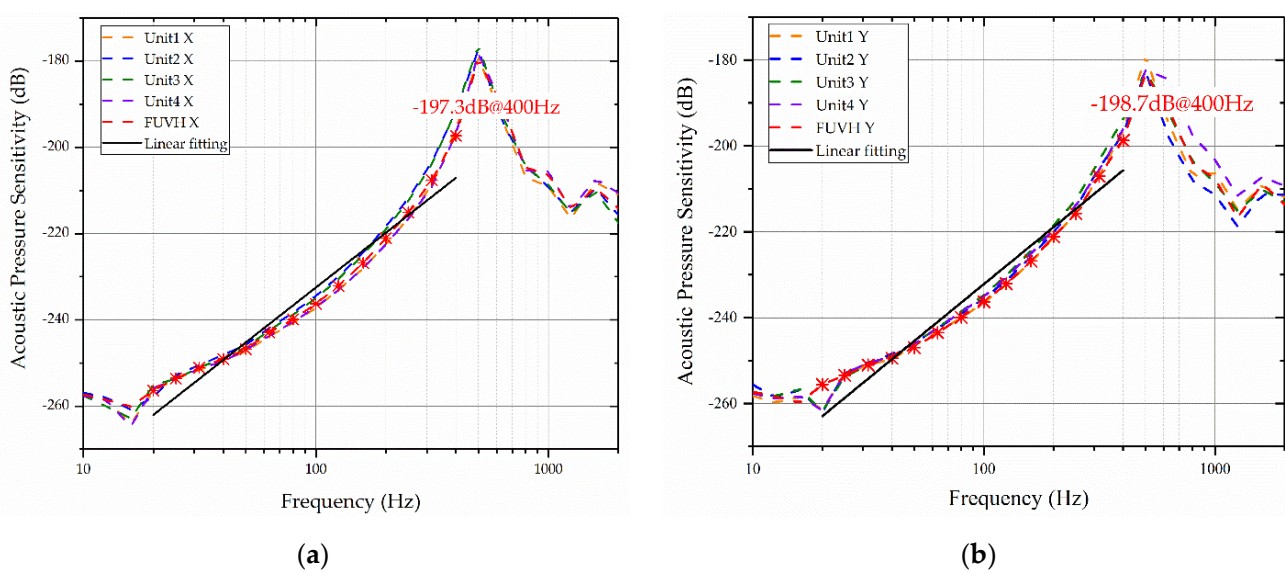

(**a**) (**b**)

**Figure 13.** The sensitivity test results: (**a**) the acoustic pressure sensitivity of the X channel is −197.3 dB@400 Hz; (**b**) the acoustic pressure sensitivity of the Y channel is −198.7 dB@400 Hz.

According to the test results, the FUVH and its four units have a resonance peak of 500 Hz. However, the available frequency band is taken at 2/3 of the resonance peak [28]. In the region of 20–400 Hz, the sensitivity curve has good linearity. As for the four units, the sensitivity curves overlap approximately. At the same frequency point, the maximum difference between each unit's sensitivity is 5.7 dB. As for the FUVH, the sensitivity coincides with the average value of the four units, and the maximum difference is 2 dB. As can be seen, the FUVH has good performance in its operating band, with a sensitivity of $-197.3$ dB@400 Hz and better linearity at low frequencies.

### 4.2. Directivity Pattern Test

In the directivity test, a signal generator generates a signal, and a power amplifier amplifies it. Then, it makes the emission transducer emit sound while the revolver rotates. In this way, the FUVH can pick up the sound signal at all angles and output a voltage signal captured by a data acquisition card. We conducted directivity pattern tests on X and Y channels of FUVH and its four units. The test frequency was 315 Hz, and the data were normalized by [34]:

$$L = 20\lg\left(\frac{e_\theta}{e_{max}}\right) + |K_d| \tag{28}$$

where $e_\theta$ is the output voltage value of the FUVH in any direction, $e_{max}$ is the maximum value of the FUVH output voltage, and the concave point depth $K_d$ expression of directivity is:

$$K_d = 20\lg\left(\frac{e_{max}}{e_{min}}\right) \tag{29}$$

where $e_{min}$ is the minimum value of the FUVH output voltage.

We plotted the normalized graph of the tested data in polar coordinates. We could obtain the directivity plot at the 315 Hz frequency point, the results of which are shown in Figure 14.

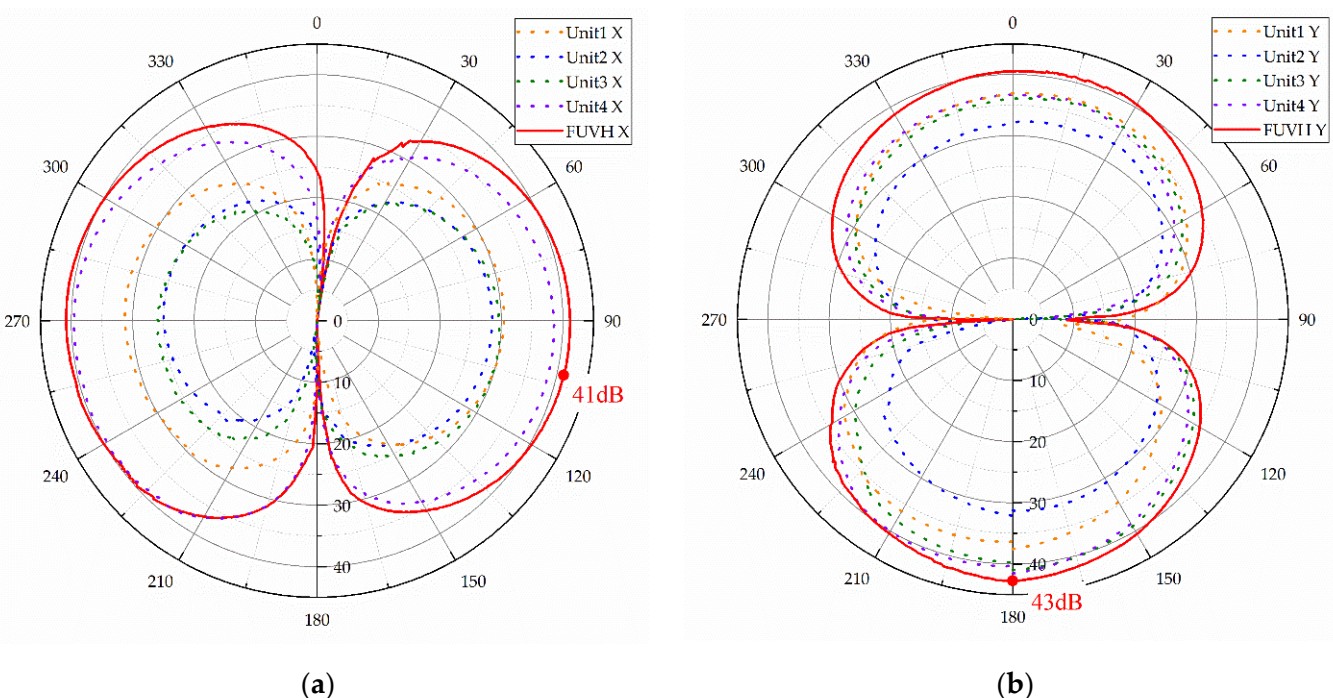

**Figure 14.** The results of the directivity pattern test: (**a**) the concave point depth of the X channel is 41 dB; (**b**) the concave point depth of the Y channel is 43 dB.

According to the test results, although the directivity of the four units of this FUVH has an "8" shape, some units have "small and large circles" in their directivity curves. This

is thought to be caused by artificial factors in the cilia integration process [35]. However, because the FUVH can average the output signals of each unit, the strengths and weaknesses are offset. As a result, the FUVH takes on a better "8" shape in the directivity curve, and the concave point depth reaches 43 dB, which is much more significant than the depth of its four units.

### 4.3. Self-Noise Test

Finally, we tested the self-noise of the FUVH and its four units. The most common method of self-noise testing is to connect the output of the sensor and its pre-circuit to a spectrum analyzer for testing [36]. The self-noise test must ensure that the environment is quiet, so the experiment was carried out during a quiet night. We kept the FUVH elastically suspended from the revolver, and submerged it in the water of the calibration tube. Then, the sensor and the pre-circuit were powered up so that the sensor was in working condition. The output was connected to the spectrum analyzer while setting the relevant parameters. The spectrum analyzer at this point shows the power spectral density of the frequency-domain noise voltage after the Fourier transform of the time-domain noise voltage signal, expressed as:

$$\mathrm{F}(w) = \int_{-\infty}^{\infty} V_{N\_out}(\tau) e^{jw\tau} \mathrm{d}\tau \tag{30}$$

where *F(w)* is the self-noise power spectral density, and the results are taken as logarithmic. The test results are shown in Figure 15, where industrial frequency and multiplier interference occur at 50 and 100 Hz.

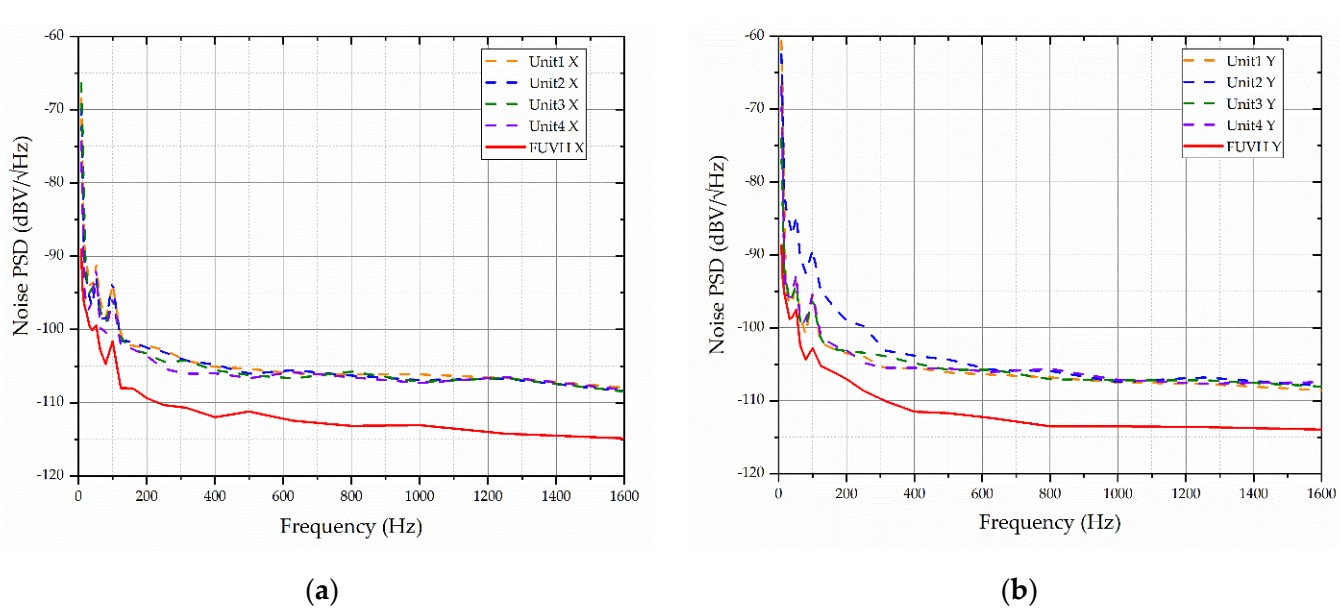

(**a**)          (**b**)

**Figure 15.** The results of the self-noise test: (**a**) the noise PSD of the X channel and (**b**) the noise PSD of the Y channel.

According to the test results, the white noise spectrum of the FUVH self-noise is $-111$ dBV/$\sqrt{\text{Hz}}$. In contrast, the white noise spectrum of its four units of self-noise is $-105$ dBV/$\sqrt{\text{Hz}}$, reducing the white noise by 6 dB, in line with the aforementioned theoretical analysis.

### 5. Results and Analysis

The evaluation of the hydrophone self-noise requires a combination of sensitivity and the self-noise spectrum. Because the sensitivity is proportional to the output voltage value, the equivalent self-noise level excludes the effect of sensitivity on the noise magnitude. It is more of a reference for evaluating the magnitude of the sensor self-noise. The equivalent

self-noise level of the sensor characterizes the equivalent sound pressure of the sensor self-noise at the input of the sensor, which is:

$$P_N = \frac{e_N}{e_x} \frac{coskd}{sinkd_0} P_0 \qquad (31)$$

where $e_N$ is the transducer self-noise voltage, $P_0$ is the sound pressure at the standard sound pressure hydrophone, and the logarithm of the result is:

$$20\lg P_N = 20\lg(\frac{e_N}{e_x} \frac{coskd}{sinkd_0} P_0). \qquad (32)$$

Because $M_0 = e_0/P_0$, then substituting into (32), we get:

$$20\lg P_N = 20\lg e_N - 20\lg(M_0 \frac{e_x}{e_0} \frac{sinkd_0}{coskd}) \qquad (33)$$

where the value is the logarithmic sensor self-noise spectral density minus the logarithmic sensor sensitivity. Then, the result in the operating band of the sensor is shown in Figure 16.

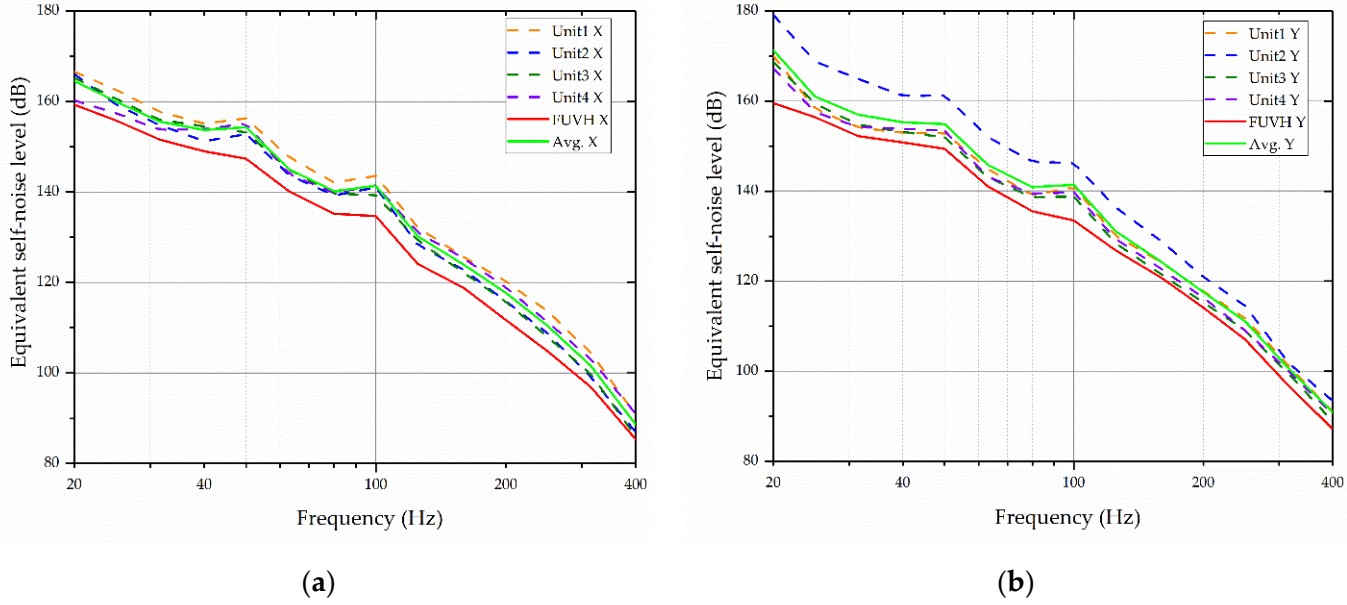

**(a)** **(b)**

**Figure 16.** The results of equivalent self-noise level: (**a**) the result of X channel and (**b**) Y channel.

As shown in Figure 16a, in X channels, the equivalent self-noise level of the FUVH is higher than the mean value of its four units, with an average difference of 5.18 dB. As shown in Figure 16b, the difference between the equivalent self-noise level of the Y channels of the FUVH and the mean value of its four units is 5.14 dB. According to the results, it is clear that the FUVH has a lower equivalent self-noise level compared to its four units.

As shown in Figure 17, we analyzed the difference between FUVH and its four units at each frequency point. The difference in the X channel is shown in Figure 17a, where most differences are around 6 dB. As shown in Figure 17b, the difference in the Y channel is mostly around 3 dB and varies greatly. The main reason for this situation is the abnormalities in the self-noise of unit 2. However, the equivalent self-noise level of the FUVH is still significantly more extensive than that of its four units.

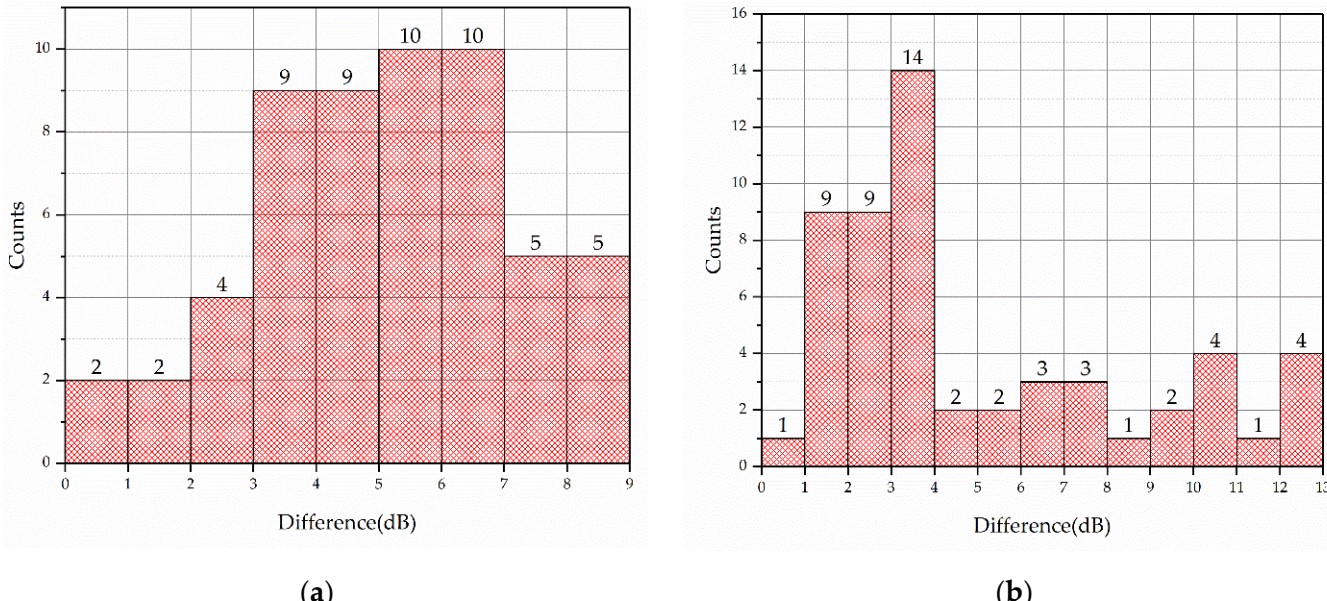

**Figure 17.** Difference statistics between FUVH and its four units: (**a**) the difference of X channel and (**b**) Y channel.

## 6. Discussion

In this study, we theoretically analyzed the noise suppression of a monolithically integrated FUVH. We fabricated a FUVH and performed calibration experiments regarding its directivity and sensitivity. The results verified that this FUVH performs well in 20–400 Hz with a sensitivity of −197.3 dB@400 Hz. We tested its self-noise level. The white noise spectrum of the FUVH is 6 dB lower than the white noise spectrum of all of its four units. The four units are considered to be four CVHs with similar performance. Finally, we analyzed the equivalent self-noise level of the FUVH in the operating band. The research validates the theoretical analysis and provides a new direction for improving the performance of the CVH to achieve a lower level of ocean ambient noise measurement. In the future, we will try to improve the process of preparing smaller structures to achieve single-point observation of more array units with a lower self-noise level. Then, we will combine the hardware circuitry and algorithms so that the CVH can achieve multi-parameter measurement, storage, and transmission.

**Author Contributions:** Conceptualization, H.T., G.L. and R.W.; methodology, H.T.; software, G.L.; validation, H.T., G.L. and H.L.; formal analysis, H.T.; investigation, G.L.; resources, W.Z.; data curation, H.T., G.L. and H.L.; writing—original draft preparation, H.T.; writing—review and editing, H.T. and G.L.; visualization, H.T.; supervision, R.W.; project administration, W.Z., G.Z., Y.Y., J.C., C.H. and L.J.; funding acquisition, W.Z. All authors have read and agreed to the published version of the manuscript.

**Funding:** This research was funded by, National Natural Science Foundation of China (Grant 52275578, 51875535, 61927807), Fundamental Research Program of Shanxi Province (20210302123027, 20210302124203), National key research and development program (2020YFC0122102), and by Shanxi "1331 Project" Key Subject Construction (1331KSC).

**Conflicts of Interest:** The authors declare no conflict of interest.

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
