# Peer review of "Research on Self-Noise Suppression of Marine Acoustic Sensor Arrays"

_remotesensing, doi:10.3390/rs14246186_

Round 1

Reviewer 1 Report

This paper research on self-noise suppression of marine acoustic sensor array. This paper is propose a solution to improve the performance of the CVH by arraying to detect environmental noise in a complex deep-water environment. We analyzed the self-noise source of the CVH and the noise suppression principle of the four-units MEMS vector hydrophone (FUVH).

Some suggestions to improve the paper are as below:

1. The language needs minor review to improve readability.

2. Also in the Introduction, it is important to enhance your motivation. Where the current surveys fail.

3. Present your contribution in the Introduction. What is your novelty actually?

4. Authors must explain in detail the introduction section.

5. For example, is Figure 1 new? Is it your compilation? 

6. What are the main limitations of the work?

7. Minor: keep the format in order to cite figures:  Figure X and not figure x.

8. Overall, the paper is very well written, and the information is interesting from the field under study. However, some important points need to be take:

a) Bullet 1) is critical, and the authors must highlight and compare with similar papers of the proposal.

9. The quality of the figures needs to be improved.

10. The conclusion must be improved and avoid redundancy

 11. References: Critical: update it.

Finally, paper needs minor revision.

Reviewer 2 Report

Acoustic sensor for underwater application is of great importance, covering a broad range of scenarios such as geological survey, ocean exploration and environmental monitoring. The performance of which is closely correlated with the vector hydrophone, therefore, study on the avenues to improve the functionality of vector hydrophone is important. In this submitted manuscript, the authors report a method to enhance the noise performance of the CVH by exploiting an array of four CVHs, named FUVH. Detailed theoretical analysis is given, in particular, the noise analysis regarding the circuitry and mechanical structure, as well as the combined equivalent noise. Experiments are carried out to characterize the sensitivity and noise performance of the FUVH, in comparison with a single CVH. Directivity measurements are also conducted in terms of X and Y channels. As a result, the proposed FUVH yields a similar sensitivity compared to a single CVH, an improved noise performance of about 5-6 dB, and an enhanced directivity pattern. Thanks for submitting this interesting work, it offers a valuable approach to optimising the performance of the vector hydrophone. However, some comments have to be addressed.

Comments:

1.      The resolution of Figure 1 is relatively low, especially, the texts.

2.      In the introduction section, lines 38 to 41, the story suddenly switched from the background to the challenge of MEMS hydrophones, it is better to add a short paragraph that covers conventional hydrophones and MEMS hydrophones.

3.      Please keep the uniformity of the gap between each line, e.g., lines 59 to 67.

4.      Equation (3) seems not correct, a bracket is missing after U0.

5.      Still about equation (3), if the assumption is made on all the resistance are equal, the result cannot reach equation (4).

6.      Please rephrase the sentence in line 130. e.g. “MEMS vector hydrophones feature silicon microstructure are advantageous in the structural dimensions and sensitivity.”

7.      About equation (8), damping is an important parameter to determine mechanical noise. However, in this particular context, it is better to give an analysis of the damping components and the impact of the structure for damping (the pillar could be the primary source).

8.      Regarding equation (9), it is kind of vague. The transfer function of the device is not difficult to obtain, at least from an experimental method. The authors already captured the responses of the noise spectrum in the range of 10-1000 Hz and a resonant peak at 500 Hz (which could be the resonant frequency of the FUVH), the approximate transfer function of the device can be garnered by using transfer function tools in Matlab, based on the experimental data.

9.      In line 303, I suggest selecting the same frequency point for minimum/maximum noise level analysis. (In the text, it was 50 Hz and 400 Hz for X-channel whereas 20 Hz and 200 Hz for Y-channel)

10.   The proposed FUVH has a working bandwidth of 20 Hz to 400 Hz, however, the working band is not a flat response but instead, an approximate linear gain. Therefore, the authors can use a linear curve fitting to quantize the linear gain, for both X-channel and Y-channel.

11.   Regarding the mechanical structure of the FUVH, the four CVHs are patterned in a square form, here leading to a question: the stress distribution for the cross beams may not be the same as for a single CVH. As for a single CVH, the cross beams are anchored to the frame, in contrast, the FUVH uses a large cross-like structure to form a mutual anchor for the four CVHs, resulting in a unbalaenced stress distribution for the cross beams. Will this be a problem for practical application?

12.   Another extended thinking: temperature flcutuations may also casue errors and uncertainty, in particular, for a cross-coupled mechanical structure. What is the FUVH temperature response compared to a single CVH? If the four CVHs within the FUVH have a different temperature coefficient, will this affect the functionality?

Round 2

Reviewer 2 Report

The authors have addressed the comments. Thank you very much for the effort.